# Meta-analysis of variable-temperature PCR technique performance for diagnosising *Schistosoma japonicum* infections in humans in endemic areas

**Meng-Tao Sun**[1], **Man-Man Gu**[1], **Jie-Ying Zhang**[1], **Qiu-Fu Yu**[1], **Poppy H. L. Lamberton**[2], **Da-Bing Lu**[1]*

**1** Department of Epidemiology and Statistics, School of Public Health, Soochow University, Suzhou, China, **2** Institute of Biodiversity, Animal Health and Comparative Medicine and Wellcome Centre for Integrative Parasitology, University of Glasgow, Glasgow, United Kingdom

* Ludabing@suda.edu.cn

## Abstract

**Data Availability Statement:** All relevant data are within the manuscript and its Supporting Information files.

### Background

As China is moving onto schistosomiasis elimination/eradication, diagnostic methods with both high sensitivity and specificity for *Schistosoma japonicum* infections in humans are urgently needed. Microscopic identification of eggs in stool is proven to have poor sensitivity in low endemic regions, and antibody tests are unable to distinguish between current and previous infections. Polymerase chain reaction (PCR) technologies for the detection of parasite DNA have been theoretically assumed to show high diagnostic sensitivity and specificity. However, the reported performance of PCR for detecting *S. japonicum* infection varied greatly among studies. Therefore, we performed a meta-analysis to evaluate the overall diagnostic performance of variable-temperature PCR technologies, based on stool or blood, for detecting *S. japonicum* infections in humans from endemic areas.

### Methods

We searched literatures in eight electronic databases, published up to 20 January 2021. The heterogeneity and publication bias of included studies were assessed statistically. The risk of bias and applicability of each eligible study were assessed using the Quality Assessment of Diagnostic Accuracy Studies 2 tool (QUADAS-2). The bivariate mixed-effects model was applied to obtain the summary estimates of diagnostic performance. The hierarchical summary receiver operating characteristic (HSROC) curve was applied to visually display the results. Subgroup analyses and multivariate regression were performed to explore the source of heterogeneity. This research was performed following the Preferred Reporting Items for Systematic Reviews and Meta-Analysis (PRISMA) guidelines and was registered prospectively in PROSPERO (CRD42021233165).

**Funding:** The authors M-TS, M-MG, J-YZ, Q-FY and D-BL and/or their research are currently funded by the National Science Foundation of China (to DL, No. 81971957). PHLL is funded by a European Research Council Starting Grant (680088 SCHISTO_PERSIST), the Wellcome Trust (grant number 204820/Z/16/Z) the Engineering and Physical Sciences Research Council (EP/T003618/1 and EP/R01437X/1), and the Medical Research Council (MR/P025447/1). The funders had no role in study design, data collection and analyses, decision to publish, or preparation of the manuscript.

**Competing interests:** The authors have declared that no competing interests exist.

## Results

A total of 2791 papers were retrieved. After assessing for duplications and eligibility a total of thirteen publications were retained for inclusion. These included eligible data from 4268 participants across sixteen studies. High heterogeneity existed among studies, but no publication bias was found. The pooled analyses of PCR data from all included studies resulted in a sensitivity of 0.91 (95% CI: 0.83 to 0.96), specificity of 0.85 (95% CI: 0.65 to 0.94), positive likelihood ratio of 5.90 (95% CI: 2.40 to 14.60), negative likelihood ratio of 0.10 (95% CI: 0.05 to 0.20) and a diagnostics odds ratio of 58 (95% CI: 19 to 179). Case-control studies showed significantly better performances for PCR diagnostics than cross-sectional studies. This was further evidenced by multivariate analyses. The four types of PCR approaches identified (conventional PCR, qPCR, Droplet digital PCR and nested PCR) differed significantly, with nested PCRs showing the best performance.

## Conclusions

Variable-temperature PCR has a satisfactory performance for diagnosing *S. japonicum* infections in humans in endemic areas. More high quality studies on *S. japonicum* diagnostic techniques, especially in low endemic areas and for the detection of dual-sex and single-sex infections are required. These will likely need to optimise a nested PCR alongside a highly sensitive gene target. They will contribute to successfully monitoring endemic areas as they move towards the WHO 2030 targets, as well as ultimately helping areas to achieve these goals.

## Author summary

*Schistosoma japonicum* is a parasite that can cause serious intestinal schistosomiasis. The infection is mainly diagnosed by parasitological and immunological methods, such as Kato-Katz test and indirect hemagglutination assay. However, both of these are not sensitive enough to accurately assess schistosomiasis elimination/eradication. PCR assays, detecting parasite DNA, are theoretically a highly suitable, sensitive and specific, alternative. However, reported performance varies greatly among studies. Therefore, we performed this meta-analysis (PROSPERO, registered No. CRD42021233165) to analyze and summarize the data from relevant studies of variable-temperature PCR for the diagnosis of *S. japonicum* infections, using stool or blood samples from humans in endemic areas. We retrieved a total of 13 eligible articles including data from 4268 participants across 16 studies. There was high heterogeneity among studies, but no publication bias was found. Analyses revealed that PCR techniques had a satisfactory performance for diagnosing *S. japonicum* infections in humans in endemic areas, with both high sensitivity and specificity. Further research on *S. japonicum* diagnostic techniques, especially in low endemicity areas and for detection of dual-sex and single-sex infections are required. These may be best using highly sensitive gene targets in nested PCR reactions.

## Introduction

Schistosomiasis is the second most important human parasitic disease in the world with approximately 240 million people infected and more than 700 million people living at risk of

infection [1]. The loss in disability-adjusted life years caused by this disease ranked third among a series of over 20 neglected tropical diseases [2,3]. The majority of human schistosomiasis is caused by three main schistosome species: *Schistosoma mansoni*, *Schistosoma haematobium and Schistosoma japonicum*. Schistosomiasis occurrs mainly in Africa and Asia, with a low prevalence in South America and the Middle East [4,5].

China and the Republic of the Philippines are the only two counties now endemic with *S. japonicum*, which is regarded to cause the most serious pathology and disease due to its highest egg output [6]. *S. japonicum* remains a major public health problem and in China has been set as one of the four top priorities for communicable disease control by the central government [7–9]. After implementing effective prevention and control measures for nearly 70 years, great progress has been achieved with a significant reduction in *S. japonicum* prevalence, intensity and associated morbidity in many endemic areas [10]. In 2014, China proposed a two-stage road map for schistosomiasis control and elimination: to achieve transmission interruption by 2020 and then to eliminate the disease at the country level by 2030 [11,12]. The government is making a considerable effort to keep the transmission of schistosomiasis interrupted region by region [13–16]. Whilst moving towards these goals of elimination, and beyond during post intervention surveillance, diagnostics that are highly sensitive and specific will be required [17–19]. These diagnostics could be critical as elimination/eradication assessment requires confirmation that no new infection cases have occurred. Any inaccurate diagnoses would make a wrong estimation of the disease prevalence and, depending on criteria cut offs, may then affect the final elimination status. However, currently there is no standardized highly sensitive and specific diagnostic technology available for *S. japonicum* infections.

At present, there are three main groups of diagnostic approaches for schistosome infections in humans: parasitological, immunological, and DNA-based methods. Parasitological assays, such as Kato-Katz detecting schistosome eggs in stool under a microscope, are recommended by the World Health Organization (WHO) for *S. japonicum* [20]. However, Kato-Katz have been proven to have a low sensitivity in people with low infection intensities and in areas with low prevalence [21]. Immunological tests can be divided into two categories: antigen tests and antibody tests. Whilst there is a highly specific antigen test available for use in the field for *S. mansoni*, there are no point of care antigen tests available for *S. japonicum*. Antibody tests are now commonly used for *S. japonicum* [22,23], but they are unable to distinguish current from previous infections [19,24,25], which make it impossible to assess active transmission and monitor for eradication/elimination. PCR-based molecular methods, to detect parasite DNA, have recently been developed and applied to the diagnosis of a broad number of infections including *Schistosoma* spp, and have primarily shown high diagnostic values [26–28]. Generally, there are four main kinds of PCR methods used to diagnose *Schistosoma* infections, namely conventional PCR, nested PCR, real-time PCR, and droplet digital PCR. Conventional PCR uses a pair of primers to amplify the target sequence millions of times. For nested PCR, a pair of primers is used to amplify the target sequence to increase the amount of template and then a pair of inner primers is used to amplify the target gene fragment [29]. Conventional and nested PCRs give a positive or negative response, but are not quantitative. Real-time PCR can quantify the copies of the target sequence in samples and is often called qPCR [30]. Droplet digital PCR is developed on the basis of water-oil emulsion droplet technology, which casuses the sample to fractionate into 1000s of droplets and the PCR reactions occur within each droplet, with the proportion of droplets that have PCR amplicon in also resulting in a quantitative measure [31,32].

As a range of protocols, and/or different biological samples, have been used for PCR diagnoses of schistosome infections, the reported diagnostic performances or estimates of such techniques have therefore also varied greatly [33]. Whilst many studies find PCR to be highly

specific and sensitive, PCR diagnostic accuracies have been reported to be considerably lower than the commonly used indirect hemagglutination assay [34]. Therefore, we performed a meta-analysis of data from studies using PCRs on DNA extracted from human serum or stools, to evaluate the overall diagnostic value of variable-temperature PCR technology for *S. japonicum* infections in terms of comprehensive indexes, when compared to the conventional Kato-Katz method.

## Methods

### Study registration

This research was performed according to the Preferred Reporting Items for Systematic Reviews and Meta-Analysis (PRISMA) guidelines (S1 Table) [35] and was registered prospectively in PROSPERO (CRD42021233165) [36].

### Searching strategy and selection criteria

This meta-analysis assessed diagnostic accuracy of the current assays based on variable-temperature PCR for the detection of *S. japonicum* in humans. We searched relevant literatures up to 20 January 2021 from electronic databases, including China National Knowledge Infrastructure (CNKI), WanFang Database, Chinese Scientific Journal Databases (VIP), PubMed, Elsevier Science Direct, ProQuest, Whiley Online Library and Web of Science. The words of "ribenxuexichong", "PCR" and "jiance" (pinyin in Chinese) were used as keywords to search Chinese databases. "*Schistosoma*", "PCR" and "detection" were used to search English databases. In order to expand the scope of the search, relevant references of the included literatures were also searched.

Literature selection criteria were that the paper included: (1) humans as subjects; (2) blood/ serum or stool as the biological samples; (3) sample sizes over 10; (4) PCR techniques limited to variable-temperature nucleic acid amplification; (5) parasitological tests (i.e. Kato-Katz or hatching test) as the reference standard; (6) both sensitivity and specificity reported, or data available so that they could be calculated. Literatures that did not meet any of the above criteria were excluded.

### Data extraction process and quality assessment

Two reviewers (MS and MG) screened the papers and performed data extraction separately, and any disagreements were resolved through discussion with the third reviewer (JZ). The following information was extracted from each eligible publication: first author; year of publication; type of evaluation; design type (case-control or cross-sectional design); reference standard; sample size; biological sample (stool or blood/serum); index test (conventional, nested, real-time or droplet digital PCR); human population; setting (endemic/non-endemic area); and the gene which the PCR targeted. Papers evaluating more than one molecular method or using more than one target gene were considered as different studies. The two reviewers independently assessed the risk of bias and applicability of each eligible study using the Quality Assessment of Diagnostic Accuracy Studies 2 tool (QUADAS-2) [37]. When there were inconsistent evaluation opinions, they would discuss with the third reviewer and made a final decision.

### Statistical analysis

Statistical analyses were performed using Review Manager 5.4 [38], Stata 15.0 [39] and R 4.0.4 [40] software. Risk of bias and applicability concern summaries and graphs for all included studies were automatically analyzed in Review Manager 5.4. The heterogeneity was assessed by

Q test and I-square test statistically. If the I-square statistic was more than 50%, and *P* was less than 0.1, then it would be considered that there was substantial heterogeneity among the studies [41]. Spearman correlation coefficients of logit (true positive rate) and logit (false positive rate) were calculated to assess if there was a threshold effect [42]. The Deek's funnel plots were used to assess the publication bias and a P value of >0.1 represented no publication bias [43]. The bivariate boxplot was employed to assess the distribution properties of sensitivity versus specificity and for identifying possible outliers [44].

The bivariate mixed-effects model [45], taking heterogeneity and threshold effect into account, was applied to obtain the summary estimates of sensitivity, specificity, positive likelihood ratio (PLR), negative likelihood ratio (NLR) and diagnostic odds ratio (DOR) by using MIDAS in STATA 15.0. A reliable diagnostic assay is suggested to have a PLR greater than 10 and a NLR less than 0.1 [46,47]. Hierarchical summary receiver operating characteristic (HSROC) curves [48] were created and applied to visually display the pooled sensitivity, specificity and their 95% confidence intervals (CIs). 95% prediction regions within each study, were denoted as a dot in the forest plots and HSROCs. The correlation between sensitivity and specificity among studies was analyzed to justify the bivariate model used. Subgroup analyses and multivariate meta-regression were performed within R to explore the source of heterogeneity and investigate the main factors influencing diagnostic accuracy.

## Results

### Literature search

Fig 1 summarizes the study selection process. We retrieved articles in eight databases: CNKI (813 related records), WanFang (707), VIP (324), PubMed (185), Elsevier Science Direct (271), ProQuest (211), Wiley Online Library (51) and Web of Science (229). Eight further records were identified through reference lists. A total of 1791 records were retained for comparison after duplicates had been removed. Of these, 1591 records were excluded due to the absence of *S. japonicum*. After reading the full texts, 187 records were deleted: one due to duplicate data, one due to having no specificity data, one with a sample size of less than ten, five with no variable-temperature PCR used and 179 with no human blood or stool samples used.

Eventually, 13 articles [33,49–60] with a total of 4268 participants were included in this meta-analysis. Three articles reported two studies regarding PCR. Xu J et al. 2011 [58] examined the diagnostic accuracy of both conventional PCR and nested PCR. Kosala G. Weerakoon et al. 2017 [33] used both blood and stool samples for their droplet digital PCR, and Yi Mu et al. 2020 [55] used two molecular markers (sja-miR-2b-5p and sja-miR-2c-5p) in their real-time PCR. Therefore, a total of 16 studies, with eleven based on serum and five on stool samples, were included. 14 studies were conducted in *S. japonicum* endemic areas of China and 4 studies were conducted in *S. japonicum* endemic areas of the Republic of the Philippines. Out of the 16 studies, six evaluated conventional PCR, three nested PCR, five real-time PCR, and two droplet digital PCR. Key details of the included studies are shown in Tables 1 and 2. Quality assessment is demonstrated in Supplement file (S1 Fig). The high-risk items were mainly Patient Selection and Index Test, in particular, seven articles were based on a case-control design and some of them did not give a clear description of whether the participants were included at random. Applicability Concerns were assessed as low risk because all the publications showed that the information which they provided met the QUADAS-2 standard.

### Heterogeneity analysis

There was no threshold effect, as indicated by the spearman coefficient ($r_s$ = 0.311, P = 0.242). When comparing paired sensitivity and specificity of all studies (Fig 2), point values were not

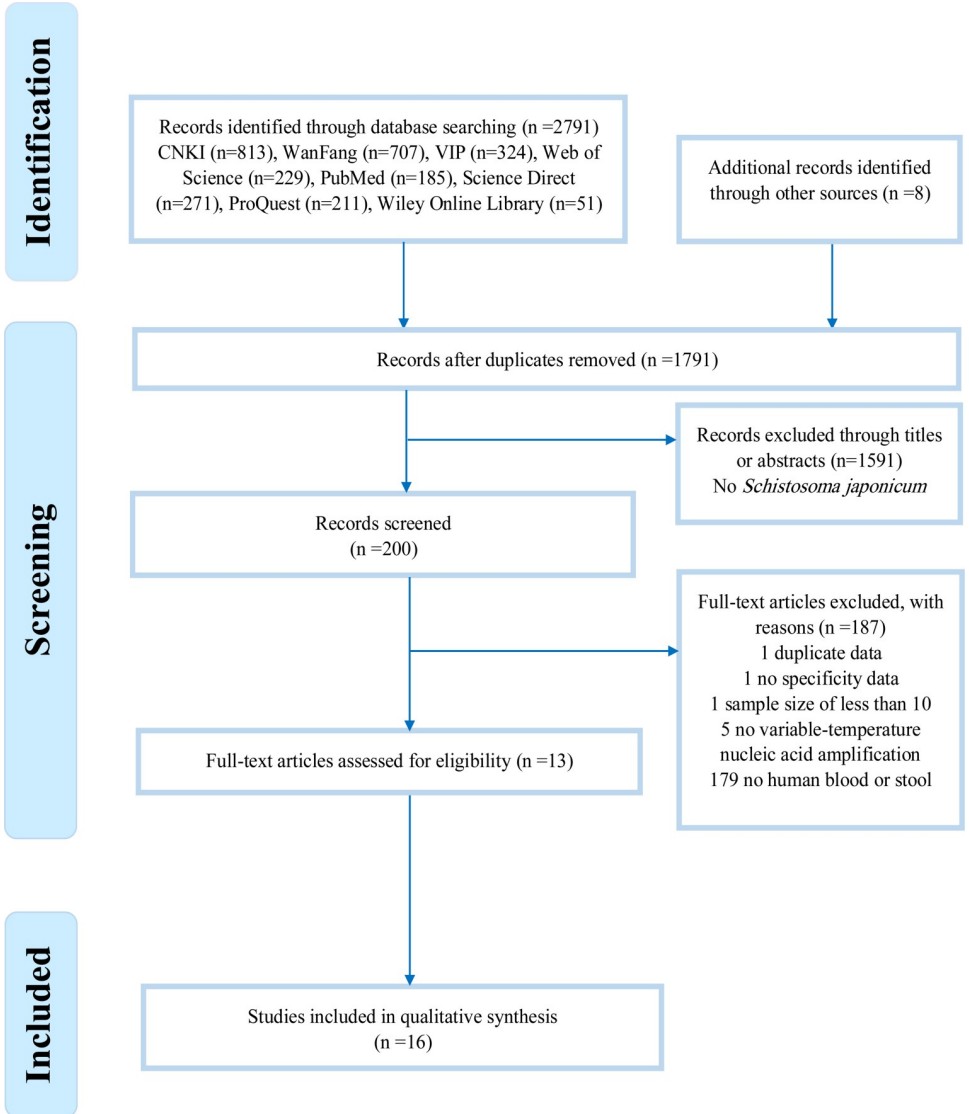

**Fig 1. Flow chart of studies on S. japonicum infections detected by PCR tests.** The chart shows numbers of titles and studies reviewed in preparation of this meta-analysis. N, the number of studies included in each stage of the process.

in a straight line. The I-squared values of sensitivity and specificity were respectively 97.30 and 99.14 (both P<0.05), suggesting an existence of substantial heterogeneity among studies. The bivariate boxplot (Fig 3) shows that no studies fell outside the 95% CIs. Deek's funnel plot (Fig 4) shows that studies were roughly equally distributed on each side of the regression line with no obvious publication bias (b = 24.09, P = 0.13).

## Pooling diagnostic accuracy

Across all studies and tests there was a negative correlation between sensitivity and specificity (r = -0.36, P = 0.17). The bivariable mixed-effects model was used to perform the analyses for the overall diagnostic accuracy and subgroup estimates. As seen in Table 3, the pooled accuracy of all 16 studies resulted in a sensitivity of 0.91 (95% CI: 0.83 to 0.96), specificity of 0.85 (95% CI: 0.65 to 0.94), PLR of 5.90 (95% CI: 2.40 to 14.60), NLR of 0.10 (95% CI: 0.05 to 0.20)

**Table 1. Characteristics of the eligible publications included in the analysis.**

| Study | Sample size | Design | Biological sample | Index test | Population | Setting | Target gene |
|---|---|---|---|---|---|---|---|
| Fung 2012 [49] | 106 | Cross-sectional | Stool | Conventional PCR | All ages | Endemic area in Sichuan, China | SjR2 |
| Gordon 2015 [50] | 560 | Cross-sectional | Stool | Real-time PCR | All ages | Endemic area in Laoang and Palapag municipalities, Northern Samar, the Philippines | NADH 1 |
| Guan 2013 [51] | 71 | Case-control | Serum | Real-time PCR | - | Endemic area in Hunan, China | SjR2 |
| Guo 2012 [52] | 94 | Case-control | Serum | Nested PCR | - | Endemic areas in Hunan and Suzhou, Jiangsu, China | SjCHGCS19 |
| Kato-Hayashia 2015 [53] | 23 | Cross-sectional | Serum | Conventional PCR | All ages | Endemic area in Sorsogon, the Philippines | COI |
| Lier 2009 [54] | 1106 | Cross-sectional | Stool | Real-time PCR | - | Endemic area in the Yangtze River in Tongling, Anhui, China | NADH 1 |
| Mu 2020[a/b] [55] | 78 | Cross-sectional | Serum | Real-time PCR | All ages | Endemic area in Laoang and Palapag, Northern Samar, the Philippines | Sja-miR-2b-5p/ Sja-miR-2c-5p |
| Wang 2009 [56] | 75 | Case-control | Stool | Conventional PCR | - | Endemic area in Anqing, Anhui and non-endemic area in Hanshan, Anhui China | 14-3-3 protein mRNA |
| Weerakoon 2017[a/b] [33] | 412 | Cross-sectional | Serum /Stool | Droplet digital PCR | All ages | Endemic area in Laoang and Palapag municipalities, Northern Samar, the Philippines | NADH 1 |
| Xu 2010 [57] | 50 | Case-control | Serum | Conventional PCR | - | Endemic area in Hunan and Suzhou, Jiangsu, China | SjR2 |
| Xu 2011[a/b] [58] | 215 | Case-control | Serum | Conventional PCR/ Nested PCR | - | Endemic areas in Hunan and Wuxi, Jiangsu, China | SjR2 |
| Xu 2014 [59] | 1371 | Cross-sectional | Serum | Conventional PCR | All ages | Endemic area of southeastern Poyang Lake, Jiangxi, China | SjR2 |
| Zeng 2017 [60] | 107 | Case-control | Serum | Nested PCR | All ages | Endemic area in Hunan, China | SjR2 |

Note

a/b, two studies within the publication.

and DOR of 58 (95% CI: 19 to 179). A global summary of the diagnostic performance of PCRs across all of the 16 studies is presented in Fig 5.

Subgroup analyses did not reveal the possible sources of heterogeneity as I square of all pooled indexes within a majority of subgroups were more than 50% with P values <0.01. Table 3 displays the pooled estimates for each subgroup. In terms of research design, the significantly higher estimates in diagnostic accuracy were obtained for case-control studies than cross-sectional studies (Chi square = 15.72, P<0.01). No significant difference was seen between stool-based and blood-based PCR (Chi square = 0.08, P = 0.78). There was significant differences among the four types of PCR techniques (Chi square = 11.90, P<0.01) with the highest estimates of accuracy obtained for nested PCR. Their HSROC curves (with 95% CIs and 95% prediction regions) are shown in Fig 6. Because there was only two studies of droplet digital PCR, the corresponding HSROC curve cannot be drawn. The multivariate regression analyses, with research design, biological samples and PCR types as covariates, showed that research design may influence the diagnostic accuracy.

## Discussion

In this meta-analysis, we analyzed and summarized the data from relevant studies of variable-temperature PCR used for the diagnosis of *S. japonicum* infections in humans in endemic

**Table 2. Raw data from included studies.**

| Author | Tp | Fp | Fn | Tn | N |
|---|---|---|---|---|---|
| Fung 2012 [49] | 14 | 2 | 8 | 82 | 106 |
| Gordon 2015 [50] | 121 | 384 | 7 | 48 | 560 |
| Guan 2013 [51] | 39 | 0 | 2 | 30 | 71 |
| Guo 2012 [52] | 42 | 2 | 1 | 49 | 94 |
| Kato-Hayashi 2015 [53] | 1 | 9 | 0 | 13 | 23 |
| Lier 2009 [54] | 32 | 35 | 39 | 1000 | 1106 |
| Mu 2020_a [55] | 36 | 8 | 17 | 17 | 78 |
| Mu 2020_b [55] | 42 | 11 | 11 | 14 | 78 |
| Wang 2009 [56] | 31 | 0 | 0 | 44 | 75 |
| Weerakoon 2017_a [33] | 102 | 176 | 6 | 128 | 412 |
| Weerakoon 2017_b [33] | 106 | 201 | 2 | 103 | 412 |
| Xu 2010 [57] | 18 | 0 | 12 | 20 | 50 |
| Xu 2011_a [58] | 96 | 7 | 14 | 98 | 215 |
| Xu 2011_b [58] | 98 | 9 | 12 | 96 | 215 |
| Xu 2014 [59] | 71 | 394 | 3 | 903 | 1371 |
| Zeng 2017 [60] | 74 | 21 | 0 | 12 | 107 |

Tp = number of true positive samples; Fp = number of false positive samples; Fn = number of false negative samples; Tn = number of true negative samples; N = total number of samples tested.

areas, based on DNA extracted from stool or blood samples. A total of 13 articles, 16 sets of data and 4268 participants were included. There was high heterogeneity among studies, but no publication bias was found. The bivariate mixed-effect model revealed that these PCR techniques had a satisfactory performance in diagnosis for *S. japonicum* infections with both high sensitivity and specificity.

The high heterogeneity between these included studies might be due to several reasons. First, this research included two distinct types of research design, case-control and cross-sectional designs. Second, sample size of included studies varied greatly, ranging from 23 to 1371 participants. Finally, studies involved four different types of PCR approaches, seven different target genes and DNA extracted from stool or blood [33,49–60]. Considering both intra- and inter-study variability, the bivariate mixed-effect model was applied in estimating the diagnostic performance of the overall studies and it's subgroups.

Sensitivity and specificity are true performance statistics of an index test. The results from our study showed the overall diagnostic performance of PCR had an average high sensitivity of 91% and specificity of 85% with the diagnostic odds ratio of 58. This was considerably higher than the estimates reported in two previous meta-analyses [34,61], of which the former [34] included four studies only, and the latter [61] was based on various biological materials and from different host species including serum, feces, urine and saliva from humans, feces from pigs and tissues from snails. Our results suggests that these PCR techniques can be used as an effective tool for diagnosing *S. japonicum* infections in humans in both clinical practice and field surveys. The low pooled NLR of 0.10 suggested that a negative PCR result may provide enough evidence to rule out *S. japonicum* infection. However, it is noted that the pooled PLR was only 5.90, less than 10 recommended [46,47], suggesting that individuals with PCR-positive results should be further confirmed by using other diagnostic assays or repeat tests. One reason for this could also come from the period of four to fourteen weeks required for parasite DNA to disintegrate and disappear in hosts after treatment [62]. Further analyses of diagnostic

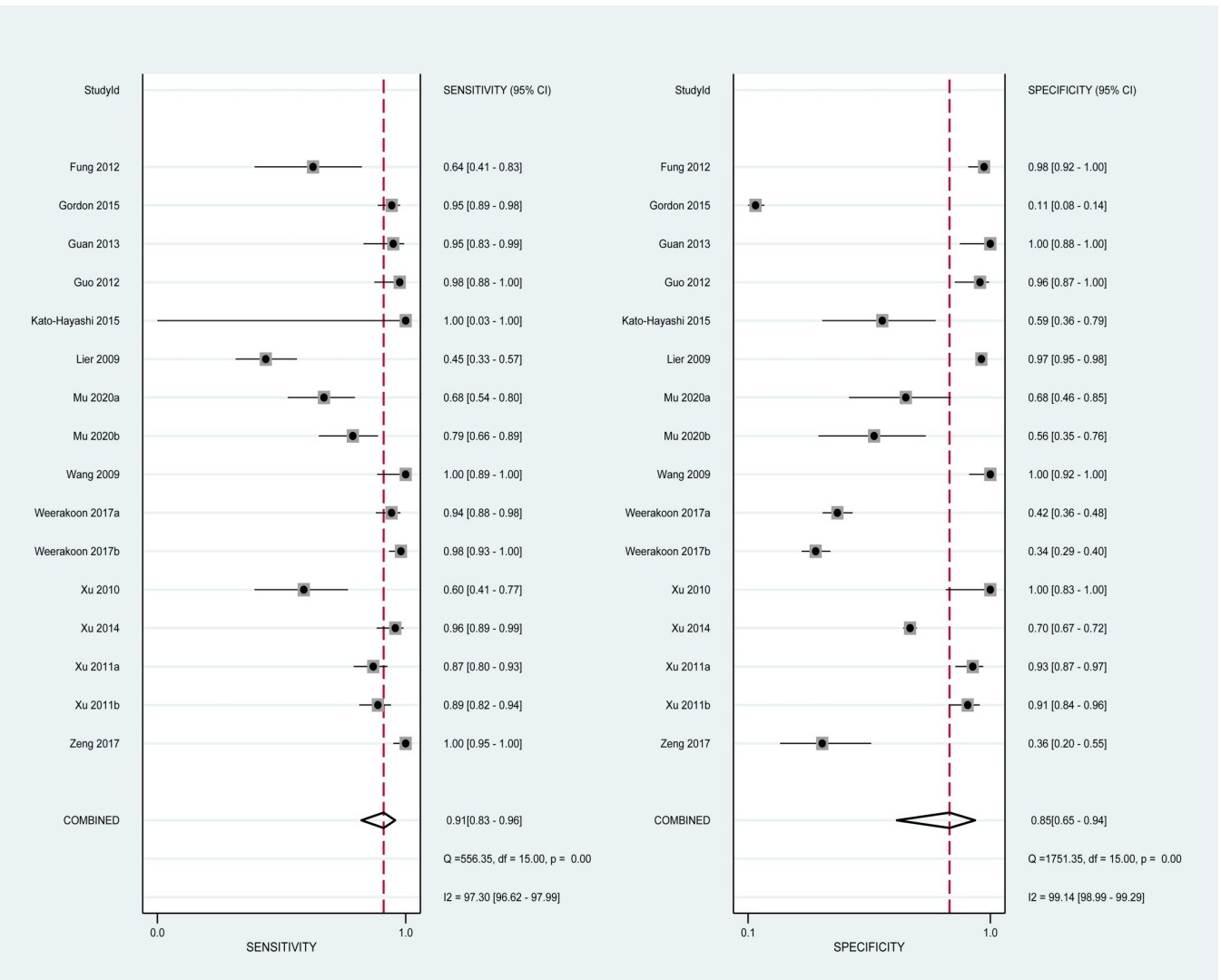

**Fig 2. Paired forests of pooled sensitivity and specificity of all included studies.**

sensitivities pre and post treatment could help to elucidate this. Furthermore this low PLR should be interpreted with caution as it is worth noting that diagnostic performance of the PCR methods in these publications, and therefore in this meta-analysis, might be underestimated due to the use of Kato-Katz as the reference standard. As Kato-Katz are known to have low sensitivity, particularly in hosts with low infection intensities, positive results by PCR for those who are Kato-Katz negative may be likely to be true positives, rather than false positives. As schistosome worms can not be directly counted, all diagnostics, whether detecting eggs, antigens, antibodies, or DNA are all acting as a proxy for worm burden. It is vital that diagnostic comparative studies moving forwards use latent class analyses given the lack of a true gold standard for *Schistosoma* infections [63,64].

Subgroup analysis was performed according to research design, biological samples and PCR types. For research design types, case-control studies reported better performances for PCR diagnostics than cross-sectional studies, as indicated by the estimates of pooled sensitivity, specificity, likelihood ratio and DOR. These better performances may partly come from a

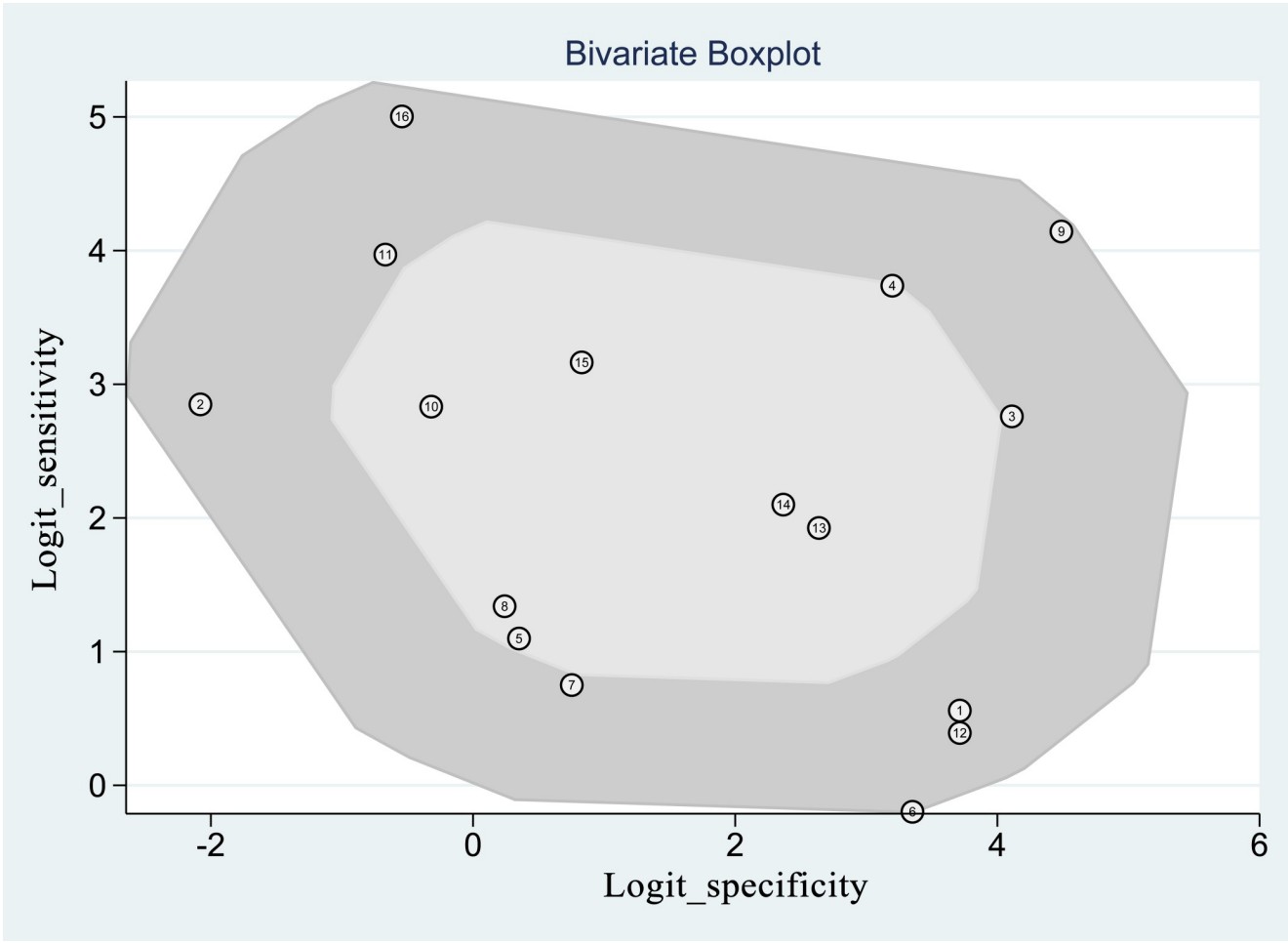

**Fig 3. Bivariate boxplots of all included studies.**

possible 'overestimation' of the diagnosis accuracy in case-control research, as previously proposed in [37,65]. In case-control studies, the participants are previously confirmed as either infected individuals (in the case group) or healthy and/or non-cases (in the control group); whereas in cross-sectional design studies, several participants truly infected with *S. japonicum* could be missed by Kato-Katz due to the low sensitivity of this reference assay [26]. Regarding biological sample type, it is generally assumed that, due to the irregular output of larval eggs in stool and the constant existence in serum of cell-free circulating DNA from schistosomes (paired or unpaired), the blood-based PCR should have better diagnostic performance than the stool-based [62]. However, we did not find any significant difference between the sample type, potentially due to the low number of studies being compared. Among the four sub groups of PCR approaches compared, the nested PCRs showed the highest diagnostic performance, indicating that further research focused on this technique may provide the optimal protocols. However, due to the overall low number of studies and comparisons, the other PCR methods should not be ruled out. Indeed qPCR and Droplet digital PCR may remain important if quantitative results are needed. Further multivariate analyses showed that the overall diagnostic performance was most affected by research design rather than by biological materials or PCR types, however as previously mentioned this may be due to the cohort selection rather than actual clinical or technical sensitivity and specificity of the specific tests.

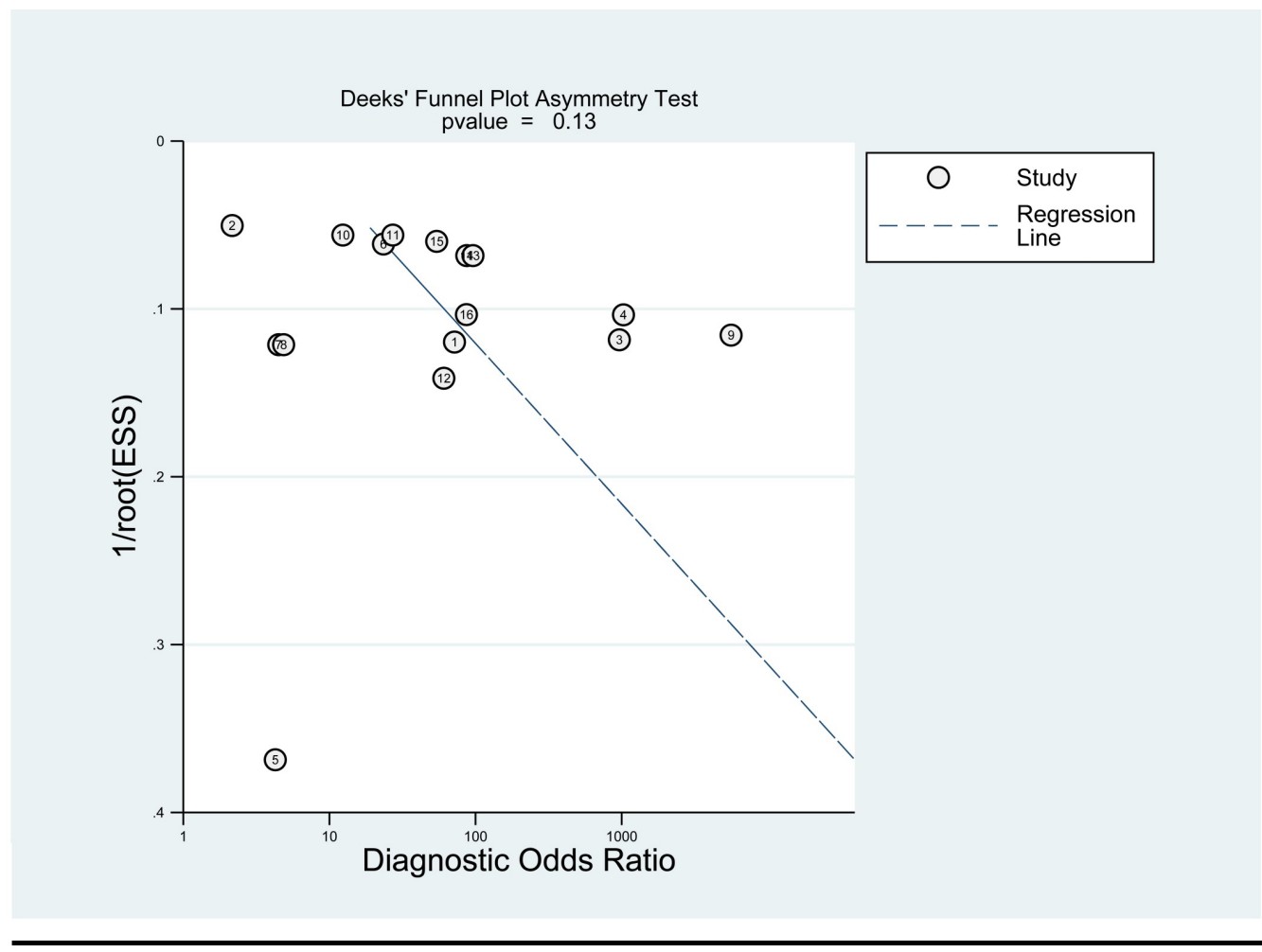

| yb | Coef. | Std. Err. | t | P>t | [95% Conf. | Interval] |
|---|---|---|---|---|---|---|
| Bias | 24.09 | 14.85 | 1.62 | 0.13 | -7.77 | 55.95 |
| Intercept | 1.70 | 1.16 | 1.46 | 0.17 | -0.80 | 4.20 |

**Fig 4. Assessment of publication bias by Deek's funnel plot asymmetry test.**

Recently, we have proposed that single-sex schistosome infections in humans could be more common than dual-sex infections [66]. Following that, we provided evidence that single-sex schistosome infections remain capable of reproducing once their opposite-sex are available [67]. For these single sex infections, the conventional parasitological assays lose function because no eggs are produced or excreted and therefore there will be no eggs in hosts' stools [68]. PCR techniques, on the other hand, detect schistosome DNA, making it possible to detect such unisexual infections [51]. Further research on the development of PCR techniques to

**Table 3. Subgroup analysis of diagnostic accuracy.**

| | Sensitivity (95% CI) | Het. (I²%, P) | Specificity (95% CI) | Het. (I²%, P) | PLR (95% CI) | Het. (I²%, P) | NLR (95% CI) | Het. (I²%, P) | DOR (95% CI) | Het. (I²/%, P) |
|---|---|---|---|---|---|---|---|---|---|---|
| **Overall** | 0.91 (0.83, 0.96) | 97.3; <0.01 | 0.85 (0.65, 0.94) | 99.1; <0.01 | 5.90 (2.40, 14.60) | 99.4; <0.01 | 0.10 (0.05, 0.20) | 97.4; <0.01 | 58 (19, 179) | 100; <0.01 |
| **Research design** | | | | | | | | | | |
| Cross-sectional | 0.87 (0.73, 0.94) | 97.4; <0.01 | 0.67 (0.39, 0.86) | 99.4; <0.01 | 2.60 (1.30, 5.00) | 99.0; <0.01 | 0.20 (0.11, 0.34) | 96.7; <0.01 | 13 (6, 28) | 100; <0.01 |
| Case-control | 0.95 (0.82, 0.99) | 92.6; <0.01 | 0.96 (0.80, 0.99) | 96.1; <0.01 | 26.0 (4.30, 158.60) | 96.6; <0.01 | 0.05 (0.01, 0.20) | 92.9; <0.01 | 512 (77, 3405) | 88.0; <0.01 |
| **Biological materials** | | | | | | | | | | |
| Stool | 0.91 (0.63, 0.98) | 99.6; <0.01 | 0.87 (0.32, 0.99) | 99.8; <0.01 | 7.20 (0.70, 72.30) | 99.9; <0.01 | 0.10 (0.02, 0.51) | 99.6; <0.01 | 69 (4, 1218) | 100; <0.01 |
| Serum | 0.92 (0.83, 0.96) | 89.9; <0.01 | 0.83 (0.63, 0.93) | 96.2; <0.01 | 5.30 (2.30, 12.30) | 95.1; <0.01 | 0.10 (0.05, 0.21) | 88.2; <0.01 | 53 (17, 165) | 100; <0.01 |
| **PCR type** | | | | | | | | | | |
| Conventional PCR | 0.86 (0.82, 0.90) | 86.5; <0.01 | 0.74 (0.72, 0.76) | 95.7; <0.01 | 11.37 (1.47, 88.06) | 97.1; <0.01 | 0.18 (0.07, 0.44) | 87.0; <0.01 | 73.76 (28.95, 187.95) | 38.2; 0.15 |
| Nested PCR | 0.94 (0.90, 0.97) | 86.0; <0.01 | 0.83 (0.77, 0.88) | 96.0; <0.01 | 7.06 (0.72, 68.88) | 97.5; <0.01 | 0.05 (0.01, 0.23) | 55.8; 0.10 | 158.62 (36.17, 695.63) | 43.1; 0.17 |
| Real-time PCR | 0.78 (0.73, 0.82) | 94.7; <0.01 | 0.72 (0.69, 0.74) | 99.7; <0.01 | 4.121 (0.74, 21.69) | 98.8; <0.01 | 0.39 (0.24, 0.62) | 74.2; <0.01 | 10.06 (2.72, 37.26) | 88.6; <0.01 |
| Droplet digital PCR | 0.96 (0.93, 0.98) | 53.9; 0.14 | 0.38 (0.34, 0.40) | 77.1; 0.04 | 1.55 (1.41, 1.70) | 47.9; 0.17 | 0.10 (0.05, 0.23) | 19.3; 0.27 | 15.24 (7.33, 31.69) | 0.0; 0.34 |

Abbreviations: Het = Heterogeneity; PLR = positive likelihood ratio; NLR = negative likelihood ratio; DOR = diagnostic odds ratio; I² = I-square; CI = confidence interval; P = P value.

detect single sex *S. japonicum* infections is warranted as we currently do not know the prevalence of such infections in humans or other definitive hosts and therefore cannot elucidate the larger implications of these as we move towards elimination of *S. japonicum*.

There are some limitations in this meta-analysis. First, out of the sixteen studies included, seven were case-control studies, in which most did not apply a blinding method which resulted in a high risk of "Patient Selection" and "Index Test". This might have overestimated the diagnostic accuracy of PCR. Second, it is impossible to analyze the influence of different target genes on diagnostic accuracy due to the small number of studies included targeting each. Thirdly, the heterogeneity was not explained fully by subgroup analyses. Fourthly, the effect of infection intensity of diagnostic accuracy could not be assessed through this meta analysis, despite it's known effect. Finally, as most published studies use Kato-Katz as the reference standard for *S. japonicum* this was part of our inclusion criteria. However, the majority of studies did not use latent class analysis to estimate the sensitivities and specificities of the different PCR tests, which may have resulted in under-estimates of specificity. However, despite this all four PCR methods were found to display high sensitivities and specificities and therefore are interpreted to be adequate for use for monitoring and evaluation in *S. japonicum* endemic areas as we move towards elimination.

## Conclusions

This meta-analysis showed that variable-temperature PCR had a satisfactory performance in diagnoses for *S. japonicum* infections. More exploration on diagnostic techniques, especially in low endemic areas for detection of *S. japonicum* dual-sex and possible single-sex infections,

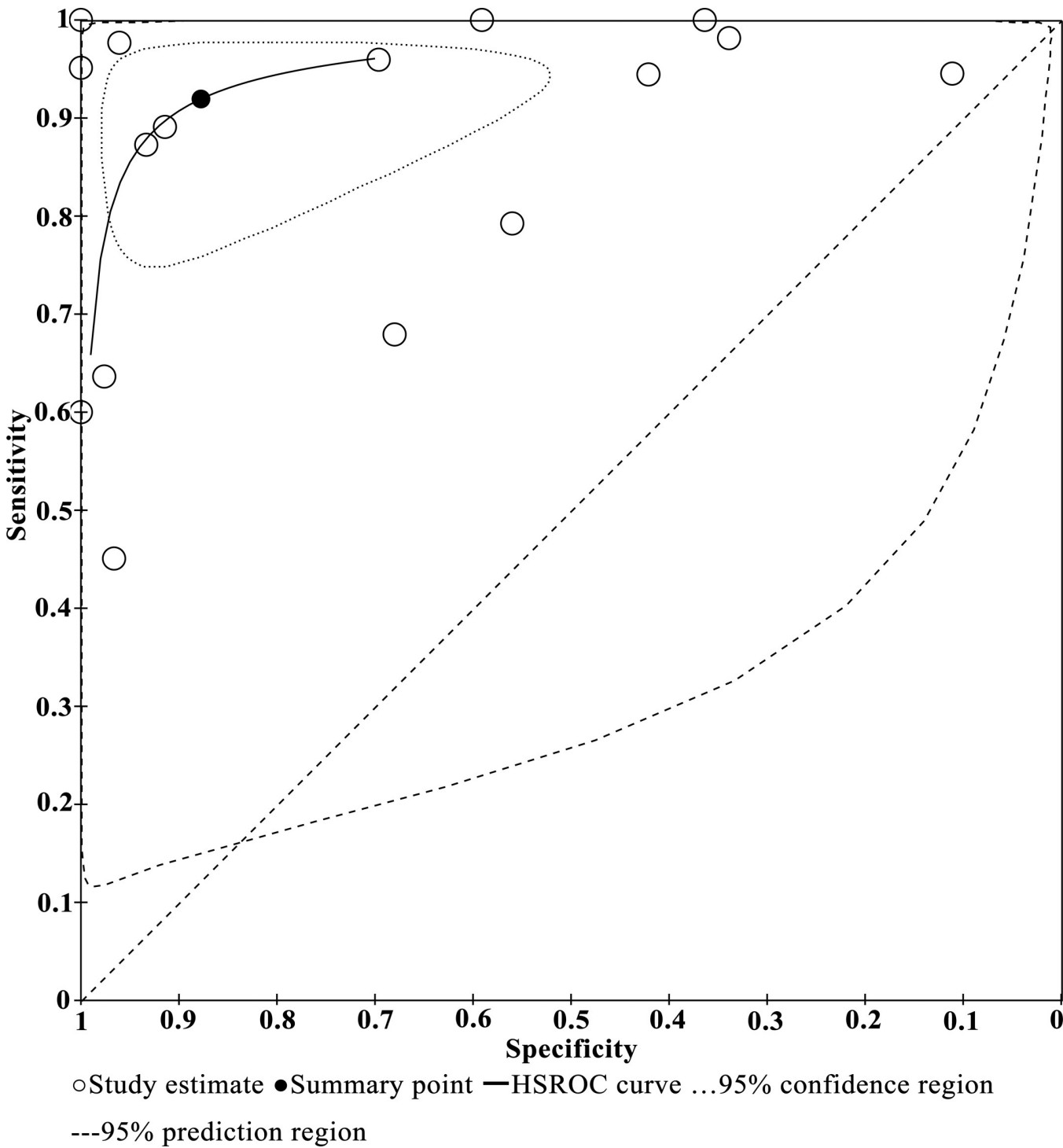

**Fig 5. HSROC curve of PCR tests used to diagnose *S. japonicum* infections.**

using high-quality studies, highly sensitive gene targets and ideally nested PCRs, is required, to help support *S. japonicum* endemic areas progress towards the WHO 2030 target, and to monitor if and when it is achieved.

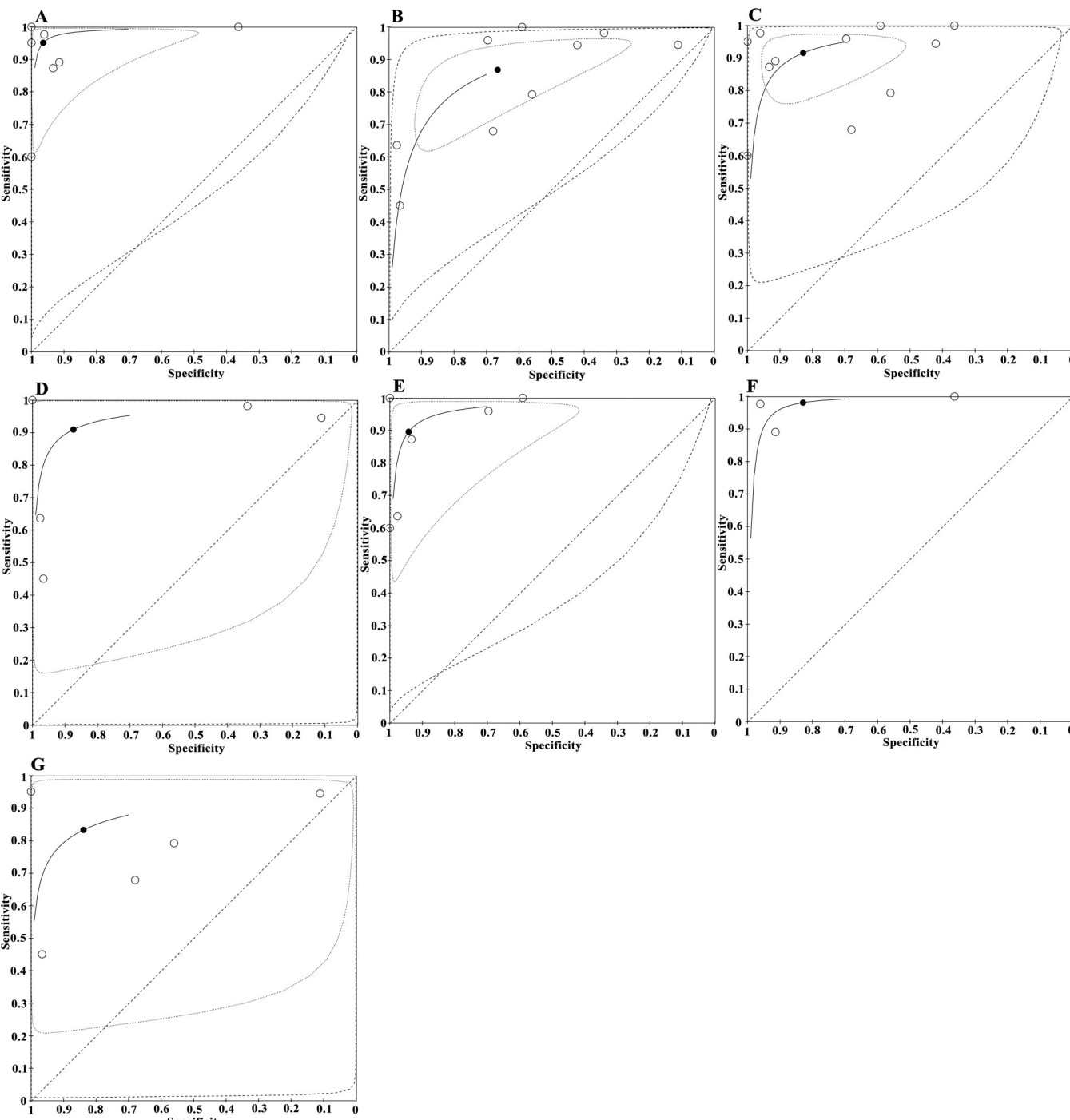

○Study estimate ●Summary point ━HSROC curve …95% confidence region ---95% prediction region

**Fig 6. HSROC curves of subgroup analyses based on research design (A and B), biological samples (C and D) and PCR method (E-G).** A) case-control design, B) cross-sectional design, C) based on blood samples, D) based on stool samples, E) conventional PCR, F) nested PCR, G) real-time PCR.

## Supporting information

**S1 Table. PRISMA checklist.**
(DOCX)

**S1 Fig.** Risk of bias and applicability concerns summary (A) and graph (B).
(TIF)

## Author Contributions

**Conceptualization:** Meng-Tao Sun, Poppy H. L. Lamberton, Da-Bing Lu.

**Data curation:** Meng-Tao Sun, Man-Man Gu.

**Formal analysis:** Meng-Tao Sun, Jie-Ying Zhang.

**Funding acquisition:** Poppy H. L. Lamberton, Da-Bing Lu.

**Methodology:** Meng-Tao Sun, Da-Bing Lu.

**Project administration:** Da-Bing Lu.

**Resources:** Meng-Tao Sun, Qiu-Fu Yu.

**Software:** Meng-Tao Sun, Man-Man Gu.

**Supervision:** Poppy H. L. Lamberton, Da-Bing Lu.

**Validation:** Meng-Tao Sun, Jie-Ying Zhang, Poppy H. L. Lamberton, Da-Bing Lu.

**Writing – original draft:** Meng-Tao Sun.

**Writing – review & editing:** Poppy H. L. Lamberton, Da-Bing Lu.

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
