## [Decision Letter · Decision Letter 0]

17 Oct 2021

Dear Dr Lu ,

Thank you very much for submitting your manuscript "Meta-analysis of variable-temperature PCR technique performance for Schistosoma japonicum infections in humans" for consideration at PLOS Neglected Tropical Diseases. As with all papers reviewed by the journal, your manuscript was reviewed by members of the editorial board and by several independent reviewers. The reviewers appreciated the attention to an important topic. Based on the reviews, we are likely to accept this manuscript for publication, providing that you modify the manuscript according to the review recommendations. 

In addition to the referees comments below, there is a need to proof read the manuscript for English grammar as there are quite a lot of grammatical errors throughout . As one example to give you an idea of what is required : eg lines 110-111 it states "However, it has been proved to be low sensitive in people...." Grammatically this should be "However, it has been proven to be of low sensitivity in people....."

Sincerely,

John Stuart Gilleard

Associate Editor

Elizabeth Carlton

Deputy Editor

In addition to the referees comments below, there is a need to proof read the manuscript for English grammar as there are quite a lot of grammatical errors throughout . As one example to give you an idea of what is required : eg lines 110-111 it states "However, it has been proved to be low sensitive in people...." Grammatically this should be "However, it has been proven to be of low sensitivity in people....."

Reviewer's Responses to Questions

**Key Review Criteria Required for Acceptance?**

**Methods**

-Are the objectives of the study clearly articulated with a clear testable hypothesis stated?

-Is the study design appropriate to address the stated objectives?

-Is the population clearly described and appropriate for the hypothesis being tested?

-Is the sample size sufficient to ensure adequate power to address the hypothesis being tested?

-Were correct statistical analysis used to support conclusions?

-Are there concerns about ethical or regulatory requirements being met?

Reviewer #1: Accepted. The authors have also searched Chinese databases looking for studies to include, and that is an advantage. The selection criteria and selection process is properly described, and so is the statistical methods used.

Reviewer #2: Yes

**Results**

-Does the analysis presented match the analysis plan?

-Are the results clearly and completely presented?

-Are the figures (Tables, Images) of sufficient quality for clarity?

Reviewer #1: Accepted. Two minor suggestions; the phrases ‘traditional PCR’ and ‘conventional PCR’ are both used in the text and figures in the meaning of non-realtime PCR. The same phrase should be used throughout, preferably ‘conventional PCR’. In Table 1 the abbreviations Tp, Fp, Fn and Tn must be explained somewhere, i.e. written out in full.

Reviewer #2: The quality of the figures (Fig 1 and 2) are not clearly visible. It need some modification.

**Conclusions**

-Are the conclusions supported by the data presented?

-Are the limitations of analysis clearly described?

-Do the authors discuss how these data can be helpful to advance our understanding of the topic under study?

-Is public health relevance addressed?

Reviewer #1: Accepted. Studies that look at diagnostic methods for Schistosoma infections has a challenge in that there is a lack of a good reference method with high sensitivity and specificity to which you can compare a ‘new’ test. When you compare a ‘new’ test (such as PCR), which might be a more accurate test, with a ‘poor’ reference method (with low sensitivity or specificity) the perceived performance of the ‘new’ test would be worse than the ‘true’ performance. Whether you for instance choose microscopy or antibodies as reference method might have a huge impact. Suggest a short mentioning/discussion of this.

The phrase ‘The common serological methods, e.g., based on the soluble antigen of eggs have also been proven to be useless.’ This might be a bit too categorical statement that is not fully supported by the reference. Suggest a slight moderation.

The authors suggest using blood-based PCR in stead of stool, mainly for practical reasons/higher compliance. This is true, but some blood-based Schistosoma PCR methods has been shown to be positive for a long time after treatment. Would that be of any consequence?

Reviewer #2: Yes

**Editorial and Data Presentation Modifications?**

Reviewer #1: The term 'highly sensitive genes' is being used. Suggest using 'highly sensitive PCR targets' or 'gene targets' as genes themselves are not 'sensitive'.

Suggest rephrasing a few sentences I find complicated or unclear:

Line 309; we have proposed that single-sex schistosome infections in humans should be more common.. Use 'could'?

Line 316: Further research on the development of the detection is wanted as we would know what prevalence of such infections in humans or other hosts could be, if such methods are available. Complicated/unclear. Rephrase?

Reviewer #2: (No Response)

**Summary and General Comments**

Reviewer #1: This study is of interest because the epidemiological situation in many of the S japonicum-endemic areas, with low prevalence and low intensity infections, makes it necessary to look into new and improved diagnostics. The results are not novel per se, but the authors do a good job by collecting and analysing the different studies, including from Chinese databases. One challenge with this meta-analysis, also commented by the authors, is the heterogenicity of the included studies in terms of study design, PCR targets/methods etc. This makes it more difficult to draw conclusions.

A reader who is interested in diagnostics for S japonicum might not be familiar with the statistical methods for meta-analysis used in this study. That you might argue is a problem for the reader, not the authors.

Reviewer #2: Reviewer report

The manuscript addresses the evaluation of molecular diagnostic technique in titled “Meta-analysis of variable -temperature PCR technique performance for Schistosoma japonicum infections in human”. Performance evaluation of sensitive diagnostic techniques such as PCR is very relevant in the area where there is a low prevalence of S. japonicum infection which is unable to detect even using the gold standard Kato-Katz techniques. The manuscript is original, relevant and well organized to be publishable with slight modifications.

Minor comments 

The title is not geographically delimited but in the result section lines, 210 & 211 indicated that all studies were conducted in schistosomiasis endemic areas of China or the Philippines. If all the data included in the analysis were from these two countries, it may need some modification of the title. 

Abstract: 

 Results: the title is about PCR technique performance. Which PCR approach performed best and which one is poorly performed? I suggest a subgroup analysis by PCR approaches.

 Conclusion: if you conduct subgroup analysis by PCR methods, your conclusions may also be modified. 

 Keywords: the keyword shall be arranged alphabetically. 

Introduction: It will be nice if you included some introductory information about some of the PCR approaches being used for the detection of S. japonicum. 

Results:

 Line 2-12-213: It is indicated that different PCR techniques were used among the 13 studies included in the meta-analysis. It will be nice if you compare the results of the different PCR approaches (traditional, two droplets digital, real-time and nested PCR). Which PCR approaches have better sensitivity, specificity, PLR, NLR and DOR? In other words, you may conduct a subgroup analysis by PCR methods. This information helps you to forward better recommendation in the types of diagnostic method shall be used in low prevalence areas. 

 Table 1: You have to add a key at the bottom of the table indicating Tp, Ep, Fn and Tn. 

 Fig 1and 2 are not visible for the readers. It needs some modifications. 

Conclusion: If you perform subgroup analysis by types of PCR techniques, your conclusion might be modified. As you recommended blood-based PCR is better than stool-based PCR approaches.

List of abbreviations: several abbreviations are available in the manuscript but lack the list of abbreviations. It will be nice if you include the list of abbreviations.

PLOS authors have the option to publish the peer review history of their article (what does this mean?). If published, this will include your full peer review and any attached files.

Reviewer #1: Yes: Tore Lier

Reviewer #2: No

Figure Files:

Data Requirements:

Reproducibility:

References

---

## [Editor Report · Decision Letter 1]

3 Jan 2022

Dear Dr Lu,

We are pleased to inform you that your manuscript 'Meta-analysis of variable-temperature PCR technique performance for diagnosing Schistosoma japonicum infections in humans in endemic areas' has been provisionally accepted for publication in PLOS Neglected Tropical Diseases.

Best regards,

John Stuart Gilleard

Associate Editor

Elizabeth Carlton

Deputy Editor

---

## [Editor Report · Acceptance letter]

12 Jan 2022

Dear Dr. Lu,

We are delighted to inform you that your manuscript, "Meta-analysis of variable-temperature PCR technique performance for diagnosising Schistosoma japonicum infections in humans in endemic areas," has been formally accepted for publication in PLOS Neglected Tropical Diseases.

Best regards,

Shaden Kamhawi

co-Editor-in-Chief

Paul Brindley

co-Editor-in-Chief
